# Soil Organic Carbon Prediction Based on Vis–NIR Spectral Classification Data Using GWPCA–FCM Algorithm

**DOI:** 10.3390/s24154930

**Published:** 2024-07-30

**Authors:** Yutong Miao, Haoyu Wang, Xiaona Huang, Kexin Liu, Qian Sun, Lingtong Meng, Dongyun Xu

**Affiliations:** 1College of Resources and Environment, Shandong Agricultural University, Tai’an 271000, China; mytshcc@163.com (Y.M.); 2023120270@sdau.edu.cn (H.W.); 2023120261@sdau.edu.cn (K.L.); 2023120267@sdau.edu.cn (Q.S.); 2021110735@sdau.edu.cn (L.M.); 2Weicheng Branch of the Weifang Natural Resources and Planning Bureau, Weifang 261000, China; hxiaona123@163.com

**Keywords:** soil spectroscopy, LUCAS, GWPCA, FCM

## Abstract

Soil visible and near–infrared reflectance spectroscopy is an effective tool for the rapid estimation of soil organic carbon (SOC). The development of spectroscopic technology has increased the application of spectral libraries for SOC research. However, the direct application of spectral libraries for SOC prediction remains challenging due to the high variability in soil types and soil–forming factors. This study aims to address this challenge by improving SOC prediction accuracy through spectral classification. We utilized the European Land Use and Cover Area frame Survey (LUCAS) large–scale spectral library and employed a geographically weighted principal component analysis (GWPCA) combined with a fuzzy c–means (FCM) clustering algorithm to classify the spectra. Subsequently, we used partial least squares regression (PLSR) and the Cubist model for SOC prediction. Additionally, we classified the soil data by land cover types and compared the classification prediction results with those obtained from spectral classification. The results showed that (1) the GWPCA–FCM–Cubist model yielded the best predictions, with an average accuracy of R^2^ = 0.83 and RPIQ = 2.95, representing improvements of 10.33% and 18.00% in R^2^ and RPIQ, respectively, compared to unclassified full sample modeling. (2) The accuracy of spectral classification modeling based on GWPCA–FCM was significantly superior to that of land cover type classification modeling. Specifically, there was a 7.64% and 14.22% improvement in R^2^ and RPIQ, respectively, under PLSR, and a 13.36% and 29.10% improvement in R^2^ and RPIQ, respectively, under Cubist. (3) Overall, the prediction accuracy of Cubist models was better than that of PLSR models. These findings indicate that the application of GWPCA and FCM clustering in conjunction with the Cubist modeling technique can significantly enhance the prediction accuracy of SOC from large–scale spectral libraries.

## 1. Introduction

Soil organic carbon (SOC) plays a key role in terrestrial ecosystems [1] and is essential for food, soil, water, and energy security [2]. The fast and accurate determination of SOC is important for global food supply and environmental protection [3]. Traditionally, SOC measurements were based on laborious soil sampling and complicated laboratory chemical analysis, which were time–consuming, costly, and environmentally unfriendly [4,5,6].

As an effective alternative, visible and near–infrared reflectance (Vis–NIR) spectra have been widely used in soil property prediction [7,8] because of their fast, convenient, and inexpensive advantages [9]. To better understand and analyze soil properties using Vis–NIR spectra, researchers developed and analyzed various spectral libraries at regional, continental, national, and global scales [5,10,11]. Previous studies, such as those by Clingensmith et al. [12] and Sarkodie et al. [13], used large–sample soil spectral libraries from the U.S. to predict SOC content through direct modeling, but the results were unsatisfactory due to the diversity of spectra and the complex relationships between soil properties and soil spectra [14].

To improve the accuracy of soil property predictions from large soil databases, researchers have explored various methods. One approach is to enhance model accuracy using deep learning and machine learning algorithms [15]. For example, Wang et al. [16] combined nonlinear modeling with memory–based learning in regional spectral libraries to predict soil pH, soil organic matter, and other properties, and achieved successful results. Another approach is based on classification principles and aims to improve model accuracy through classification modeling [17,18]. For example, Ogen et al. [19] used a spectral angle mapper algorithm, spectral gradient, and fuzzy *k*–means clusters for spectral clustering, followed by modeling. Stevens et al. [20] used the Land Use and Cover Area frame Survey (LUCAS) spectral library to predict SOC content and suggested that, in subsequent studies, researchers could improve prediction accuracy by classifying samples according to the soil type or SOC content and then locally modeling the samples. Previous research has shown that local modeling through classification yields superior results. For example, Shi et al. [10] used principal component analysis (PCA) and the fuzzy *k*–means method to predict soil organic matter for 1581 soil samples from 14 provinces in China. This combination significantly improved prediction accuracy (R^2^ = 0.899; RPD = 3.158) compared with using partial least squares regression (PLSR) alone (R^2^ = 0.697; RPD = 1.817). Liu et al. [21] established the Chinese forest soil spectral library containing 11, 213 soil samples and combined the density peaks clustering algorithm with the Cubist model. The prediction ability of SOC content (R^2^ = 0.96, RPIQ = 5.83) improved significantly compared to the traditional global PLSR modeling method (R^2^ = 0.75, RPIQ = 1.95). In these studies, researchers have proved that spectral classification using soil reflectance spectral properties significantly enhances the accuracy of model predictions [22,23].

The key to effective classification lies in how the spectral features are used to classify soil samples. Because of the multidimensionality of spectral data, data dimensionality reduction is performed prior to classification. Previous studies mainly used PCA for data dimensionality reduction [10,24]. However, PCA does not consider the spatial variation of locations, which can result in less accurate principal component extraction, especially on large scales. Our study is the first attempt to apply geographically weighted principal component analysis (GWPCA) for spectral data processing. GWPCA considers the uniqueness and spatial variation of locations, overcoming the limitations of traditional PCA in determining the weights of spatially varying indicators [25,26]. Therefore, we used GWPCA to extract principal components and reduce the dimensionality of spectral data.

Fuzzy c–means (FCM) is a clustering method based on fuzzy theory that demonstrates good adaptability by assigning membership values to each sample, thereby effectively clustering the data [27]. Previous studies have shown the excellent results of FCM in the digital mapping of taxonomic soil units and the delineation of natural soil environments [28]. However, few studies have used FCM for spectral classification to predict soil properties. In this study, we propose a spectral classification method based on GWPCA–FCM to classify and localize the modeling of large–scale soil spectral libraries in a simple, fast, and accurate manner.

Various data analysis techniques have been explored continuously to build predictive models of soil properties using Vis–NIR, such as convolutional neural networks [29,30], multiple linear regression [31,32], and PLSR [8,33]. PLSR is the most widely used linear model, and for high–dimensional multicollinearity, it is more stable and has higher prediction accuracy than traditional methods [23]. However, in spectral libraries with large–scale samples, the multivariate nature of soil spectral data and the nonlinear relationship between soil properties and spectral data [14] make it difficult for the linear model to directly explain the relationship between the spectra and soil properties [34]. Cubist is an advanced, nonparametric regression tree algorithm that can handle nonlinear relationships [35]. Peng et al. [36] found that Cubist achieved the best performance in modeling when PLSR, random forest (RF), and Cubist were used to predict soil salinity. There is no consensus on the best spectroscopic calibration method. In this study, we use the classical linear model PLSR and the machine learning method Cubist to establish SOC prediction models and compare the prediction effects for linear and nonlinear models.

Therefore, we aim to integrate classification approaches and machine learning techniques to provide an effective and accurate spectral prediction method for SOC content in large–scale regions. The main objectives are as follows: (1) to evaluate the performance of GWPCA–FCM in improving SOC spectroscopic prediction based on the LUCAS large–scale spectral library; (2) to compare the prediction accuracy of spectroscopic models based on spectra classification and land cover type classification; and (3) to determine the optimal modeling strategy for SOC prediction.

## 2. Materials and Methods

### 2.1. Study Area

We obtained data from LUCAS conducted by the Statistical Office of the European Union in 2008–2012 and used indoor soil Vis–NIR (400–2500 nm) spectra for the study [37]. The study area covered 23 Member States of the European Union, including Sweden, Spain, and the Netherlands. The terrain of the study area is diverse, with plains dominating and little relief, and land cover types such as cropland, woodland, and grassland dominating, with a total area of 4.38 million km^2^. The climate types are complex and diverse, covering 35 climate zones, most of which have temperate oceanic climates, with warm winters and cool summers, and a small annual temperature difference. A few of them have Mediterranean, temperate continental, and polar climates. The region is characterized by major European soil types, such as gray soil, brown soil, desert soil, charcoal soil, and chestnut calcium soil. The main crops grown in the region include wheat, maize, and sugar beet. See Figure 1.

### 2.2. Soil Sampling and Spectra Measurement

A total of 19,967 topsoil samples (0–20 cm) were collected for different land cover types, including cropland, woodland, shrubland, and grassland, which covered all major soil types in Europe. After the sampling locations were determined using a multi–stage stratified random sampling approach, five soil samples were taken within 2 m of the sampling points using the criss–cross method. The latitude and longitude of the sampling points were recorded using GPS during sampling.

The soil samples were air–dried and sieved (2 mm) following the protocol described by the manufacturer and the Soil Spectroscopy Group [38]. The Vis–NIR soil spectra were measured using a FOSS XDS Rapid Content Analyzer (FOSS NIR Systems Inc., Laurel, MD, USA), operating in the 400–2500 nm wavelength range with a spectral resolution of 0.5 nm.

### 2.3. Spectral Data Preprocessing

To reduce data redundancy and improve modeling efficiency, the original spectral reflectance was first resampled with a resampling interval of 10 nm. The resampled spectra were transformed to absorbance (log(1/*R*)), which can enlarge the spectral differences between samples and highlight the spectral characteristics of the soil. To remove background and noise effects, such as environmental factors, intrinsic factors of the samples’ own reflections, and the electrical noise of the spectrometer, the spectra were smoothed using the Savitzky–Golay algorithm which was fitted using a constant with a polynomial of order 2 and a window size of 15.

### 2.4. GWPCA

GWPCA aims to account for certain spatial heterogeneity in data and is one of the main methods for multivariate data analysis [39]. Unlike conventional PCA, GWPCA takes into account spatial variation in the covariance structure of the variables, whose covariances are appropriately weighted using a distance function between the target and neighboring variables:(1)$$\sum u,v=X^{T}W\left(u,v\right)X,$$
where X is an *n* × *m* matrix, *n* is the number of samples, *m* is the number of variables, and $W\left(u,v\right)$ represents the diagonal matrix of distance weights at position coordinates (u,v). At position ($u_{i},v_{i})$, GWPCA defines the local feature structure as follows:(2)$$L\left(u_{i},v_{i}\right)V\left(u_{i},v_{i}\right)L^{T}\left(u_{i},v_{i}\right)=\sum \left(u_{i},v_{i}\right),$$
where $L\left(u_{i},v_{i}\right)$ is the eigenvector matrix, which represents the loading of each independent variable for each principal component. $V\left(u_{i},v_{i}\right)$ is the diagonal matrix of eigenvalues. The score of each principal component can be expressed as
(3)$$Z\left(u_{i},v_{i}\right)=X\left(u_{i},v_{i}\right)L\left(u_{i},v_{i}\right).$$

Prior to the application of PCA, first, the data are standardized for the independent variables, and PCA is specified using the covariance matrix. The number of principal components is determined using the magnitude of the eigenvalues (eigenvalues greater than one are chosen for this study). The optimal bandwidth of the retained principal components is chosen based on the weighting function “bisquare”. The bisquare kernel function is given by
(4)$$w_{ij}=\left\{\begin{array}{c} {\left(1-{\left(\frac{d_{ij}}{b}\right)}^{2}\right)}^{2},\;if\left|d_{ij}\right|<0\\ \;\;\;\;\;0,\;\;otherwise \end{array}\right.\;$$
where the bandwidth is the geographical distance *b*, and $d_{ij}$ is the distance between the spatial locations of the *i*th and *j*th row in the data matrix. The final results of the principal component scores for each variable at each point are used as inputs to the FCM algorithm for cluster analysis.

The GWPCA was implemented in R using the “GW model” package [40].

### 2.5. Spectral Classification Methods

In this study, the FCM classification method was used to optimize the clustering results by minimizing an objective function that contained the distance between the affiliation degree and clustering centers and to provide an analytical index of the optimal number of classifications. The value function of the FCM method can be expressed as
(5)$$J_{PCM\left(U,C,X\right)}^{m}=\sum_{i=1}^{c}\sum_{j=1}^{n}\mu_{ij}^{m}d_{ij}^{2},$$
where *n* is the number of sample points, which is the number of principal components used in this study. *C* is the number of classifications; $\mu_{ij}$ indicates the degree to which $x_{j}$ belongs to $X_{i}$, and must satisfy 0 ≤ $\mu_{ij}$ ≤ 1 and $\overset{c}{\underset{i=1}{\sum }}\mu_{ij}=1$, $\forall_{j}=1,2,\ldots ,n$; and $d_{ij}^{2}$ is equal to the square of the Euclidean distance from $x_{j}$ to the center of $X_{i}$ clustering. *m* is the degree of fuzziness, which is a parameter that controls the flexibility of the algorithm. If *m* = 1, the result of FCM is hard c–means clustering, and as *m* increases, the clustering results become fuzzier.

The validity of the clustering results is evaluated using a fuzzy performance index (FPI) and normalized classification entropy (NCE):$$\mathrm{FPI}=1-\frac{c}{c-1}\left[1-\frac{\sum_{k=1}^{n}\sum_{i=1}^{c}{\left(\mu_{ik}\right)}^{2}}{n}\right]$$
(6)$$\mathrm{NCE}=\frac{n}{n-c}\left[-\frac{\sum_{k=1}^{n}\sum_{i=1}^{c}\mu_{ik}\log_{a}\left(\mu_{ik}\right)}{n}\right],\;\;$$
where *c* is the number of clusters, *n* is the number of samples, and $\mu_{ik}$ is the fuzzy affiliation degree. FPI represents the degree of separation between the *c* clusters in the data matrix and ranges from 0 to 1. The closer the value of FPI to 0, the less data are shared by the clusters, and the division of the clusters is obvious. The opposite case indicates that the division is ineffective. NCE is used to estimate the amount of decomposition of fuzzy c–partitioning. The smaller the value of NCE, the better the clustering effect.

### 2.6. Model Construction and Evaluation

In this study, PLSR and Cubist were used to model the inversion of SOC content and compare prediction accuracies. Before modeling, the sample soil spectra and corresponding SOC data were divided into a modeling dataset and validation dataset, and the Kennard–Stone (KS) algorithm was used to divide the data. The Euclidean distance between the spectral variables of the samples was computed. The modeling samples were selected uniformly in the feature space of the samples. The ratio of the number of modeling sets to the number of validation sets was 2:1. A log transformation was performed on the dependent variable SOC to make it conform to the normal distribution.

PLSR is a multivariate statistical analysis method with wide applicability that was proposed by Wold and Alban in 1983 and has been developed in recent years. It combines PCA, multiple linear regression, and typical correlation analysis into one regression model for solving the problem of multicollinearity encountered in multiple linear regression analysis. The PLSR algorithm integrates compression and regression steps and selects continuous orthogonal factors to maximize the covariance between the predictor and response variables [41], which are used to build a predictive model for predicting the values of the response variable. It has the advantages of simplicity and stability, easy qualitative interpretation, higher prediction accuracy, and is suitable for spectral analysis, which has more independent variables [21].

The Cubist model is a comprehensive decision tree–based learning algorithm that predicts or categorizes data by constructing multiple decision trees. A decision tree represents a segmented multivariate linear function that predicts the value of a variable through a series of independent variables. The basic concept is to create subsets of samples with similar attributes in the original data set when a variable is predicted through the constructed multivariate linear function [42], and then model each subset separately. The training rule is simple and effective and is suitable for solving the problem of the nonlinear relationship between SOC and predictor variables.

In this study, the root mean square error (RMSE), coefficient of determination (R^2^), and ratio of performance to inter–quartile distance (RPIQ) were used to verify the performance of the models. The higher the R^2^ and lower the RMSE, the higher the model prediction accuracy. RPIQ takes into account both the prediction error and variation in observations, thereby providing a more objective and easier approach to compare model effectiveness metrics in model validation studies. The larger the RPIQ, the better the predictive ability of the model. According to Salazar et al. [43], prediction ability can be divided into four categories based on the RPIQ value: RPIQ < 1.5 indicates that the model is very bad; if RPIQ is between 1.5 and 2.0, this indicates that the model is poor; if RPIQ is between 2.0 and 2.5, this indicates that the model is good; and RPIQ > 2.5 indicates that the model is very good.

### 2.7. Important Band Analysis

RF was used in this study for band importance analysis. It is an integrated machine learning algorithm that can aggregate ideas in addition to solving classification and regression problems [44]. It is applied gradually in feature importance analysis [45]. RF can quantify the degree of contribution of input features to the model output. The importance of variables is influenced by two main parameters: the size of the subset of input variables (mtry) and the number of trees in the forest (ntree).

## 3. Results and Discussion

### 3.1. Statistical Analysis of SOC Content

The histograms of raw SOC and log–transformed logSOC distributions are shown in Figure 2. The raw SOC data were positively skewed, and the logSOC data essentially conformed to a normal distribution, which satisfied the requirements of data analysis and prediction modeling [24]. Specifically, the SOC content ranged from 0 to 586.8 g kg^−1^, with a mean, median, and standard deviation of 50 g kg^−1^, 20.8 g kg^−1^, and 91.3 g kg^−1^, respectively. The skewness coefficient was 3.68, and the kurtosis coefficient was 13.46. The logSOC content ranged from 0.69 to 6.37 g kg^−1^, with a mean value of 3.22 g kg^−1^, standard deviation of 1.01 g kg^−1^, and skewness coefficient and kurtosis coefficient of 0.997 and 1.13, respectively.

To enable a further comparison and analysis of the differences in the distribution of SOC content, soil samples from four land cover types were divided. The distribution characteristics of grassland, cropland, shrubland, and woodland logSOC content are shown in Figure 3b. The distribution of SOC content in different land cover types showed obvious differences. The woodland SOC content distribution was the most dispersed, with a median value of 3.77 g kg^−1^ and the highest mean value of 3.90 g kg^−1^. This is because of the enhancement of the CO_2_ absorption capacity of woodland in the atmosphere [46]. The woodland samples were mainly taken from evergreen broadleaf forests and coniferous forests, which had a higher content of organic carbon than the other forests. Additionally, woodland was less affected by human influence and had a greater accumulation than depletion of organic carbon, which led to the highest content of organic carbon. This was followed by shrubland (mean = 3.49 g kg^−1^) and grassland (mean = 3.33 g kg^−1^). Cropland had the lowest mean SOC content of 2.72 g kg^−1^, which was mainly attributed to the generally higher carbon loss caused by soil disturbance that resulted from traditional farmland management compared with grassland and forest ecosystems [47].

### 3.2. Spectra Classification and SOC Content

Given the considerable volume of data in the soil spectral library and the issue of multiplicity correlation between bands, all soil samples were subjected to data dimensionality reduction using GWPCA. The cumulative contribution of the first four principal components exceeded 99.45%. The FCM was then used for rational classification of soil spectra and the determination of the optimal number of classifications. The data from the first four principal components were imported for FCM cluster analysis. The maximum number of iterations was 300, the convergence threshold was 0.001, and the fuzzy weighting index was 1.5 [48]. To determine the optimal number of categories, all the samples were divided into 2, 3, 4, 5, 6, 7, 8, 9, and 10 clusters. The values of FPI and NCE for each cluster are shown in Figure 4. The optimal number of clusters in this study was finally determined to be 4 by taking into consideration the matching of the land cover types and the clustering index.

The distribution characteristics of the SOC content of soil samples for the four clusters are shown in Figure 3a. After GWPCA–FCM was used, the difference in SOC content was obviously reduced, and the distribution was essentially the same. For Cluster 3, which consisted of 5145 soil samples, 4778 soil samples were mainly from four land cover types, that is, cropland (2173 samples), grassland (1111 samples), shrubland (117 samples), and woodland (1377 samples), which accounted for 93% of the soil samples for Cluster 3. The distribution of SOC content was most concentrated, with the lowest mean value of only 2.99 g kg^−1^. This is mainly because cropland soil samples account for 42% of the total number of soil samples, which is greatly influenced by cropland. Cluster 4 consisted of 4992 soil samples, mainly from four land cover types, that is, cropland (1605 samples), grassland (1235 samples), shrubland (113 samples), and woodland (1629 samples), for a total of 4582 samples, which accounted for 92% of the total soil samples for Cluster 4. The distribution of SOC was relatively scattered, with the highest mean value of 3.50 g kg^−1^. This is attributed to the fact that the woodland and grassland soil samples accounted for 57% of the total soil samples and were highly influenced by woodland and grassland. The SOC contents for Cluster 1 and Cluster 2 were comparable, with average contents of 3.15 and 3.24 g kg^−1^, respectively. This is mainly because the proportion of woodland and grassland soil samples from Cluster 2 is larger than Cluster 1.

### 3.3. Spectral Characteristics of Different Soil Types

The average spectral reflectance and its range of variation for each type of soil sample based on land cover type and GWPCA–FCM are shown in Figure 5. The spectral curves obtained by the two classification methods had basically the same morphology; however, the difference in spectral reflectance based on GWPCA–FCM was more obvious. Under the GWPCA–FCM classification, the slope of the curve was large, and the reflectance increased rapidly in the visible range. In the near–infrared region, the curve tended to flatten and the reflectance increased slowly; between 700 nm and 2500 nm, the difference in spectral reflectance gradually increased. The spectral curves had distinct absorption valleys near 1400 nm, 1900 nm, and 2200 nm, which were mainly caused by moisture, organic matter, iron oxides, and clay fractions [49]. The highest spectral reflectance was found in Cluster 3, followed by Clusters 1 and 2, and the lowest spectral reflectance was found in Cluster 4. The average spectral reflectance of Cluster1, Cluster2, Cluster3, and Cluster4 was 3.15 g kg^−1^, 3.24 g kg^−1^, 2.99 g kg^−1^, and 3.50 g kg^−1^ respectively, which showed a significant negative correlation with the increase in SOC content. The position of spectral curves decreased, and the spectral reflectance decreased.

For the land cover type, four land cover types had a similar curve shape. In the range of 400–750 nm, reflectance values increase rapidly, while they decrease slowly in the range of 800–1800 nm. Absorption features could be identified near 1400 and 1900 nm, which are assigned to soil hygroscopic water in clay minerals [50]. The highest spectral reflectance was found in cropland, followed by shrubland and grassland, and the lowest in woodland, which is consistent with the findings of Liu et al. [51]. This is because cropland soils have a lower mean SOC content (2.72 g kg^−1^) than woodland soils (3.94 g kg^−1^), shrubland soils (3.49 g kg^−1^), and grassland soils (3.33 g kg^−1^).

### 3.4. Spectral Prediction of SOC

To further explore the spectral prediction effect after GWPCA–FCM classification, the KS algorithm was used to divide the datasets of the four clusters into a modeling set and a validation set in a 2:1 ratio. The SOC spectral prediction models for the clusters were established using two methods: PLSR and Cubist. Additionally, the SOC spectral prediction models of the four land cover types were also established for comparative analysis.

The introduction of GWPCA–FCM significantly improved prediction accuracy. Specifically, the use of PLSR improved the R^2^ mean from 0.72 to 0.74 and the RPIQ mean from 2.36 to 2.43 compared with unclassified global modeling. Similar conclusions were reached by Ward et al. [24] and Liu et al. [52], who modeled spectral classification based on *k*–means clustering with PCA and found that the prediction of SOC improved, while significantly reducing the algorithm’s run time. By contrast, the accuracy of land cover type classification did not improve but declined, with the R^2^ reducing from 0.72 to 0.69 and the RPIQ reducing from 2.36 to 2.13. Using Cubist, compared to unclassified global modeling, R^2^ increased by 10.33%, RMSE decreased by 17.42%, and RPIQ increased by 18.00%. However, the accuracy of land cover type classification did not improve, with R^2^ decreasing from 0.75 to 0.73, RMSE improving from 0.33 g kg^−1^ to 0.39 g kg^−1^, and RPIQ reducing from 2.50 to 2.29. Stenberg et al. [53] noted that the prediction error of spectral models increases as the standard deviation of predicted soil properties increases. Ignoring the spatial extent and distribution of samples, the large variation in SOC content across land cover types resulted in a decrease in prediction accuracy. As shown in Figure 6 and Figure 7, after the land cover classification, the scattered points were distributed in a certain area, and the whole was more dispersed, indicating that the correlation between the predicted values and the measured values is weak. After the classification based on GWPCA–FCM, the distribution of points was more concentrated in a straight line, and the trend line was closer to a 1:1 line compared with the land cover classification, indicating that the predicted values were closer to the measured values as a whole, and the prediction effect was better.

The prediction performance of PLSR and Cubist was explored further. For GWPCA–FCM–Cubist compared with GWPCA–FCM–PLSR, the mean value of R^2^ improved from 0.74 to 0.83, the mean value of RMSE decreased from 0.48 g kg^−1^ to 0.39 g kg^−1^, and the mean value of RPIQ improved from 2.43 to 2.95, which is consistent with previous research results [23,54]. This is because the performance of PLSR is affected by multicollinearity [55]. Different data types, differences in dataset sizes, and the distribution of organic carbon content can have multiple effects on the prediction accuracy of the model. The results show that Cubist achieved higher accuracy when the spectral prediction of SOC content was performed in the context of large spatial and temporal variability, significant spatial heterogeneity, and the large data volume of the European LUCAS spectral library. See Table 1 and Table 2.

### 3.5. Important Band of SOC for Each Soil Type in RF Models

The results of the land cover classification and GWPCA–FCM spectral classification of importance bands in RF models are shown in Figure 8. In Figure 8a, the curves of the four land cover types are significantly different. For grassland, the most important bands were mainly distributed in the regions of 400–550 nm and 2210–2350 nm, with obvious peaks and valleys at 540, 1480, and 1990 nm, which may be influenced by hydroxyl vibration in the samples. For woodland, the most important bands were mainly distributed in the regions of 490–590 nm and 830–1030 nm, with distinct peaks and valleys at 920, 1660, and 1850 nm. For shrubland, the most important bands were mainly distributed in the regions of 520–610 nm and 790–940 nm, with distinct peaks and valleys at 610, 1370, and 2040 nm. For cropland, the most important bands were mainly distributed in the regions of 400–570 nm and 1800–1870 nm, with distinct peaks and valleys at 480, 980, and 1560 nm.

As shown in Figure 8b, the curves did not differ much based on GWPCA–FCM, and the distributions of higher values of feature importance, peaks, and valleys were essentially the same. The most important bands were all mainly distributed in the spectral regions of 400–600 nm and 2200–2340 nm, and obvious peaks appeared near 530, 1330, and 2030 nm. There were obvious valleys near 670, 1230, and 1970 nm, which were because of the content of, for example, organic carbon, iron oxides, and clay minerals. The features in the near–infrared band were mainly caused by the multiplicative or combined frequency absorption of the molecular vibrations of C–H, N–H, C–O, O–H, and Fe–O groups in minerals [53,54,56]. The higher importance value of the features in the 400–600 nm band was mainly affected by soil carbon and iron oxides. The main moisture absorption bands were near the 1400 nm and 1950 nm bands. Moisture absorbed electromagnetic waves in this band, and the combined frequency jump of stretching vibration and corner vibration of O–H functional groups in water molecules formed the largest absorption coefficient in the near–infrared region. The absorption band of Al–OH clay minerals mainly existed near 2200 nm in the synchrotron region, and an organic matter–related C–H characteristic peak existed near 2300 nm [52].

## 4. Conclusions

In this study, we used PLSR and Cubist models to compare SOC prediction accuracy based on full sample data, land cover classification data, and spectral classification data by GWPCA–FCM in a large spectral library. The main conclusions are as follows: (1) The prediction accuracy of the GWPCA–FCM classification model was significantly higher than that of the unclassified global model and the land cover type classification model. This approach enhanced the accuracy of SOC predictions for large spectral libraries. (2) Among the modeling approaches, Cubist was found to be superior to PLSR, with the GWPCA–FCM–Cubist model achieving the optimal prediction results. This research underscores the potential of integrating advanced data reduction and classification techniques with robust modeling algorithms to improve the precision of SOC content prediction on a large scale.

## Figures and Tables

**Figure 1 sensors-24-04930-f001:**
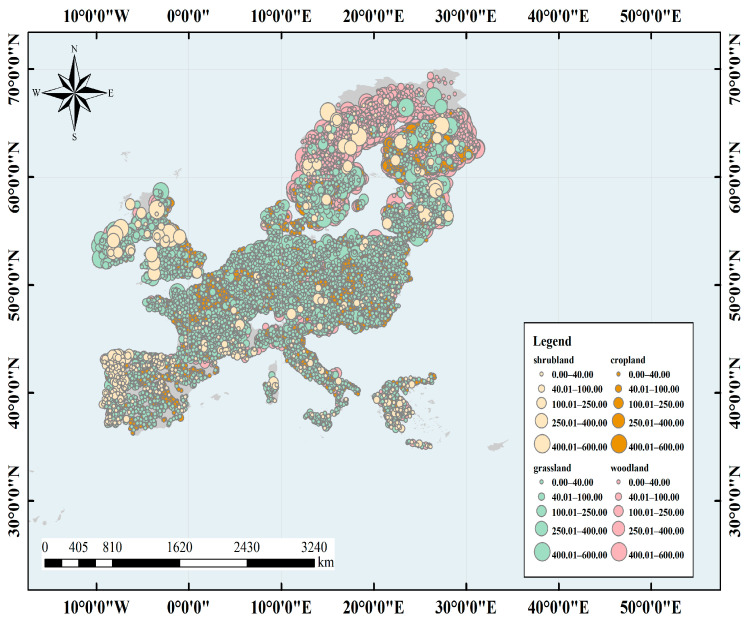
Location of soil samples from the Land Use and Cover Area frame Survey (LUCAS) soil spectral library. The color indicates the corresponding land cover type.

**Figure 2 sensors-24-04930-f002:**
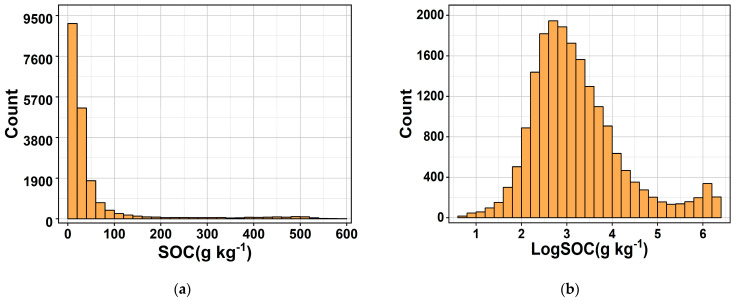
Histogram of soil organic carbon content SOC (**a**) and log–transformed logSOC (**b**).

**Figure 3 sensors-24-04930-f003:**
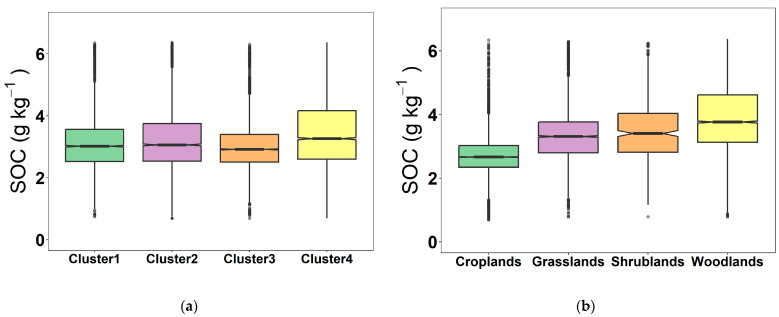
Box line plot of SOC content for land cover type (**a**) and spectral cluster (**b**).

**Figure 4 sensors-24-04930-f004:**
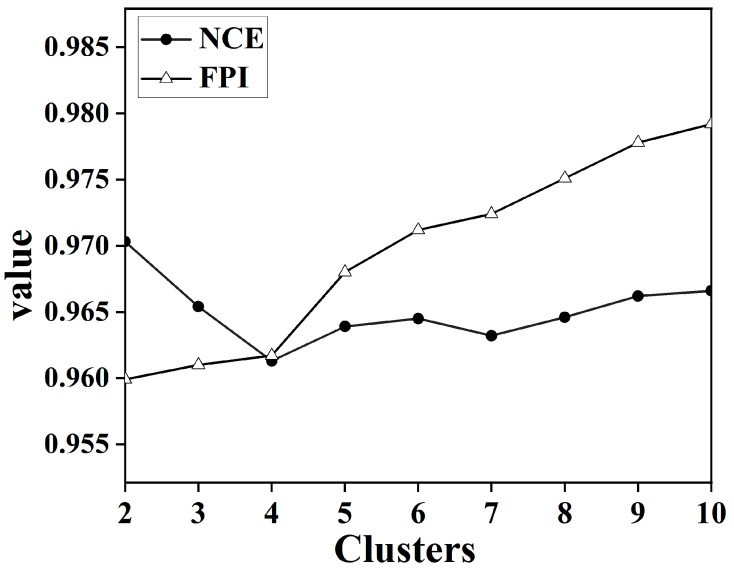
Plot of NCE and FPI value versus the number of clusters.

**Figure 5 sensors-24-04930-f005:**
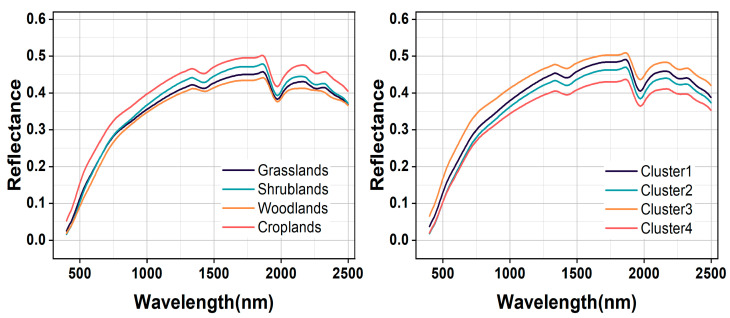
Comparison of mean soil reflectance curves based on the land cover type and GWPCA–FCM classification.

**Figure 6 sensors-24-04930-f006:**
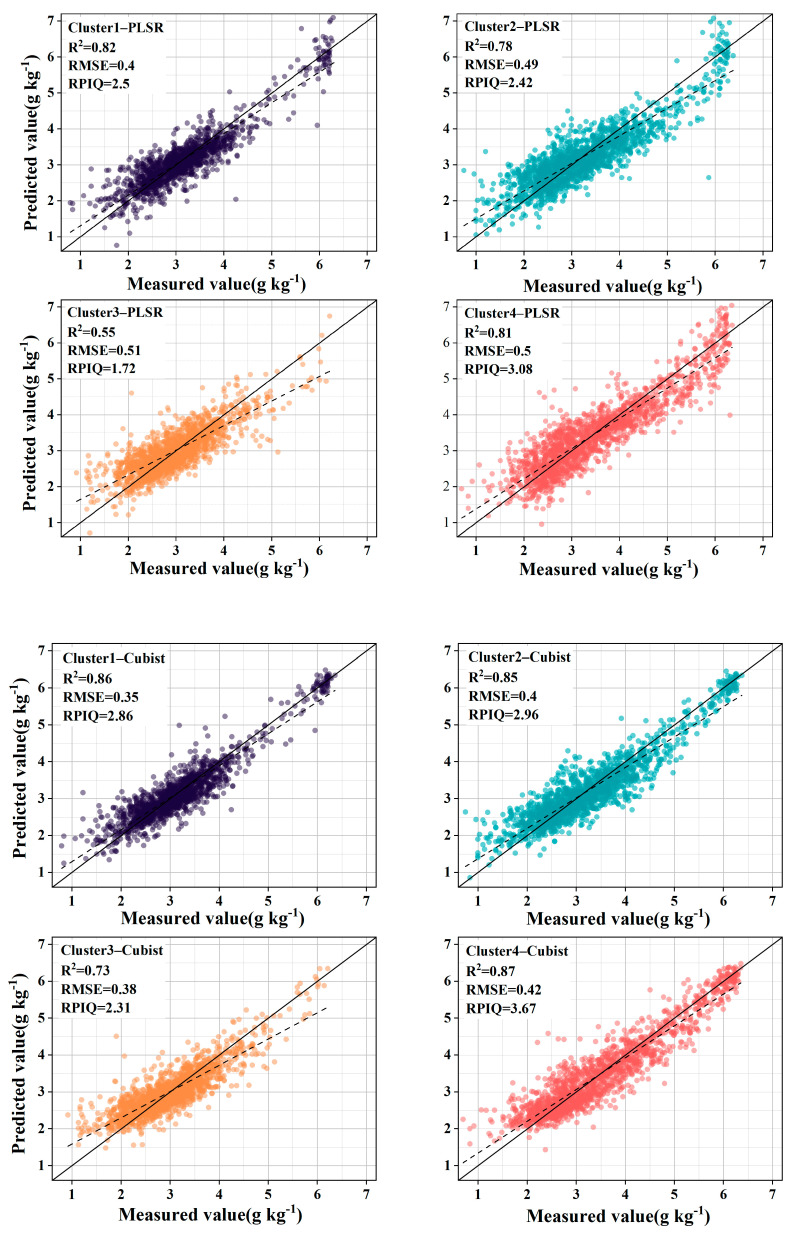
Observed vs. predicted SOC values of the validation samples for the GWPCA–FCM approaches.

**Figure 7 sensors-24-04930-f007:**
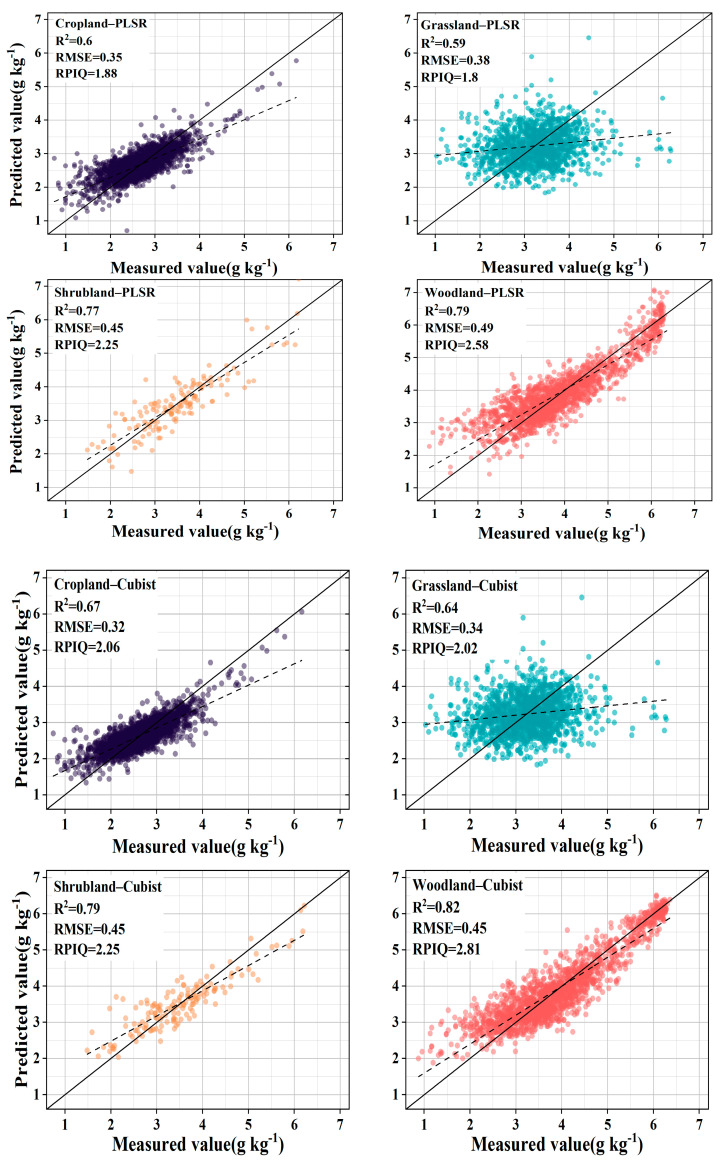
Observed vs. predicted SOC values of the validation samples for land cover type approaches.

**Figure 8 sensors-24-04930-f008:**
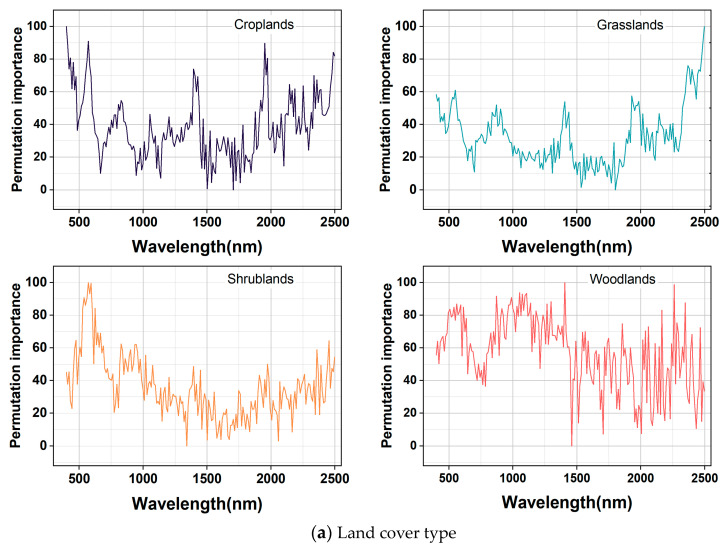
Important band diagram in RF models.

**Table 1 sensors-24-04930-t001:** PLSR and Cubist predictions of SOC for different spectral clusters.

Subsets	N	Cubist			PLSR		
		R^2^	RMSE	RPIQ	R^2^	RMSE	RPIQ
All	18,921	0.75	0.33	2.5	0.72	0.35	2.36
Cluster1	3870	0.86	0.35	2.86	0.82	0.4	2.5
Cluster2	4914	0.85	0.4	2.96	0.78	0.49	2.42
Cluster3	5145	0.73	0.38	2.31	0.55	0.51	1.72
Cluster4	4992	0.87	0.42	3.67	0.81	0.5	3.08
Mean	4730	0.83	0.39	2.95	0.74	0.48	2.43

**Table 2 sensors-24-04930-t002:** PLSR and Cubist predictions of SOC for different land cover types.

Subsets	N	Cubist			PLSR		
		R^2^	RMSE	RPIQ	R^2^	RMSE	RPIQ
All	18,921	0.75	0.33	2.5	0.72	0.35	2.36
Cropland	7476	0.67	0.32	2.06	0.6	0.35	1.88
Grassland	4200	0.64	0.34	2.02	0.59	0.38	1.8
Shrubland	443	0.79	0.45	2.25	0.77	0.45	2.25
Woodland	5218	0.82	0.45	2.81	0.79	0.49	2.58
Mean	4334	0.73	0.39	2.29	0.69	0.42	2.13

## Data Availability

The LUCAS topsoil dataset used in this work was made available by the European Commission through the European Soil Data Centre managed by the Joint Research Centre (JRC), https://esdac.jrc.ec.europa.eu/content/lucas-2009-topsoil-data, accessed on 1 July 2024.

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
