# Peer review of "Soil Organic Carbon Prediction Based on Vis–NIR Spectral Classification Data Using GWPCA–FCM Algorithm"

_sensors, 2024, doi:10.3390/s24154930_

Round 1
Reviewer 1 Report
Comments and Suggestions for Authors
The complexity of soil types and soil-forming factors in the spectral library renders direct application for predictive modeling less than ideal. In this article, the effect of SOC prediction based on spectral classification using LUCAS spectral library, combined with GWPC and FCM algorithm is explored. PLSR and Cubist modeling are used to analyze the prediction results in comparison with the prediction results for different land cover types. Multiple experiments were performed to validate the efficacy of the model. The study presented in the paper possesses certain theoretical implications.
This paper has several problems:
1. The abstract of the paper does not sufficiently condense the issue at hand and needs to clearly define the problem the paper aims to solve.
2. The introduction didn’t provide sufficient background and include all relevant references, which also leads to the manuscript not being very clear about the problem to be solved.
3. It is recommended to supplement the paper with experiments on the model's performance in different method for data set partitioning, along with results and analysis, to support the conclusions drawn in the paper.
4. Please provide a detailed analysis of Figure 6 and Figure 7.
Comments on the Quality of English Language1. This manuscript format does not meet journal formatting requirements.
2. The purpose and setup of the experiment was not described in enough detail and the results of the experiment were not analysed enough.
3. Further polish is needed for the English expression.
Reviewer 2 Report
Comments and Suggestions for Authors
The manuscript contains some useful content, but does not emphasize the novelty of the study, and it is recommended to highlight descriptions such as method upgrades or specific contributions to the analytical objectives; The title of the manuscript should be the focus of the research, but the full text does not focus on it. There are many problems in the language of the manuscript, including word errors, grammatical errors, and confusing sentence structure, so it is recommended to seek professional corrections. In addition, the innovations of the study should be discussed in depth to enhance the relevance of the study. The specific comments are as follows:
1. In the part of model validation, it is recommended to add Relative Percent Deviation (RPD) to enhance the accuracy of model accuracy evaluation.
2. In Chapter 3.4, is the data involved in the content consistent with the data in the table? Further verification is recommended.
3. The relationship between soil organic carbon content and spectral data is not described in detail in the manuscript, so it is recommended to add this part and focus on the analysis of spectral differences in soils collected from different land use types.
4. The conclusion should mention what specific methods improve the stability of prediction; However, it is not clear whether the research method and model are applicable to other soil components, so the stability and generality of soil property prediction are not improved.
Reviewer 3 Report
Comments and Suggestions for Authors The research in this study mainly stressed the issues about the Machine Learning Algorithms combined with geographically weighted principal component analysis (GWPCA) on the vis-NIR spectral library from the LUCAS. This study showed an innovative method in using LUCAS library and can provide some useful advice in practical usefulness. The conclusions are consistent with the evidence and arguments presented and the references are also appropriate. However, the discussion should be making more comparisons especially in the aspects that why the methods of GWPCA can provided the best results. Additional details: The abstract should be checked through because the sentences are illogical in lines 16,19,22,25. Figure should be made again and it was small to see.
1.Vis-NIR spectra is not equal to Hyperspectral, please correct them in the whole manuscript.
2. Figure shoud be make bigger than this manuscript
3. Figure 8, the importance of what model? please make the details in the title of figure and the corresponding paragragh.
5. line 510, the Rossel.R.A.V?
6 . Please check the reference,there were some mistakes
Comments on the Quality of English LanguageThe manuscript were be organised very well, the quality of English language were very well.
Round 2
Reviewer 2 Report
Comments and Suggestions for Authors
The author has revised the paper according to the suggestions.Please pay attention to the following three points:
1. It is suggested that the application of spectral classification methods in this study be highlighted;
2. What is the significance of the study of spectral curves and spectral data for four land cover type in section 3.3?
3. Check that the vocabulary is used correctly: for example, should "improved" be changed to "decreased" in line 366?
